# A Roman provincial city and its contamination legacy from artisanal and daily-life activities

Genevieve Holdridge[1,2], Søren M. Kristiansen[1,2]*, Gry H. Barfod[1,2], Tim C. Kinnaird[3], Achim Lichtenberger[4], Jesper Olsen[1,5], Bente Philippsen[1,5], Rubina Raja[1,6], Ian Simpson[7]

1 Centre for Urban Network Evolutions (UrbNet), Aarhus University, Højbjerg, Højbjerg, Denmark, 2 Department of Geoscience, Aarhus University, Aarhus C, Denmark, 3 School of Earth and Environmental Sciences, University of St Andrews, St Andrews, Scotland, United Kingdom, 4 Institute for Classical Archaeology and Christian Archaeology/Archaeological Museum, Münster University, Münster, Germany, 5 Aarhus AMS Centre, Department of Physics and Astronomy, Aarhus C, Denmark, 6 Department of Classical Studies, School of Culture and Society, Aarhus University, Aarhus C, Denmark, 7 Biological and Environmental Sciences, University of Stirling, Stirling, Scotland, United Kingdom

* smk@geo.au.dk

**Data Availability Statement:** All relevant data are within the manuscript and its S1 Dataset, S1 File, and S1, S2 Tables.

## Abstract

Roman metal use and related extraction activities resulted in heavy metal pollution and contamination, in particular of Pb near ancient mines and harbors, as well as producing a global atmospheric impact. New evidence from ancient Gerasa (Jerash), Jordan, suggests that small-scale but intense Roman, Byzantine and Umayyad period urban, artisanal, and everyday site activities contributed to substantial heavy metal contamination of the city and its hinterland wadi, even though no metal mining took place and hardly any lead water pipes were used. Distribution of heavy metal contaminants, especially Pb, observed in the urban soils and sediments within this ancient city and its hinterland wadi resulted from aeolian, fluvial, cultural and post-depositional processes. These represent the contamination pathways of an ancient city-hinterland setting and reflect long-term anthropogenic legacies at local and regional scales beginning in the Roman period. Thus, urban use and re-use of heavy metal sources should be factored into understanding historical global-scale contaminant distributions.

## Introduction

One of the markers of modern industrial and urban environmental impacts is heavy metal contamination in soils, sediments, air and water at local, regional and global scales [1]. However, modern contamination is not the only source with a global traceable impact. Roman industrial mining and smelting activities resulted in a global atmospheric impact observed in European peat bogs and Arctic and Alpine ice cores [2, 3]. At local and regional levels, moderate to high levels of Cu and Pb heavy metals from Roman copper, lead and silver mining and smelting activities have resulted in detectable atmospheric contamination within peats and lakes in Europe [3, 4]. In addition, high Pb and Cu pollution and contamination of colluvial and fluvial deposits are also evident around mines in Europe and the Middle East [5–7]. In

**Funding:** This research was undertaken within the framework of the Danish-German Jerash Northwest Quarter Project of the Universities of Aarhus and Münster. This project was supported by the Carlsberg Foundation (R.R., www. carlsbergfondet.dk, Grant CF14-0467), Danish National Research Foundation (R.R., www.dg.dk, Grant 119), the Deutsche Forchungsgemeinschaft (A.L., https://www.dfg.de/, grant nos LI978/4-1 and LI978/4-2), the Deutscher Palästinaverein (A.L., https://www.palaestina-verein.de/), the Danish EliteForsk Award (R.R., https://ufm.dk/forskning-og-innovation/forskningsformidling/eliteforsk, grant 4094-00077B), and H. P. Hjerl Hansens Mindefondet for Dansk Palæstinaforskning (RR). The funders had no role in the study design, data collection and analysis, decision to publish, or preparation of the manuscript.

**Competing interests:** The authors have declared that no competing interests exist.

relation to urban centers, the use of Pb and Cu in Roman cities has resulted in moderate to high levels of contamination observed in adjacent city harbors, reaching levels similar to modern industrialization levels [8]. This heavy metal pollution is attributed to the use of Pb pipes in water supply networks in Roman cities [8, 9].

Although contamination associated with mining activities and lead pipes is evident, there is little understanding of how small-scale but common and intense Roman urban, artisanal and everyday site activities resulted in heavy metal contamination of ancient urban centers and their hinterlands. These are significant omissions when discussing regional and global contamination loadings resulting from Roman activity and, more generally, in the understanding of humans as creators of contamination pathways in the early Anthropocene. Furthermore, there have been few considerations of how later Byzantine and Umayyad urban activities may have contributed to environmental contamination loadings or how they compare to the Roman period [10].

Our aim is to define the contamination legacy of an ancient middle-sized city, Gerasa/Jerash in Jordan, one of the famous classical sites of antiquity, which flourished from the early Roman period until the middle of the 8th century CE, when it was hit by a devastating earthquake (Fig 1). We identify the small-scale but common and intense Roman, Byzantine and Umayyad workshop activities in an urban quarter that may have contributed to heavy metal contamination of local soils and sediments and indicate the contamination pathways that existed both within an ancient city and between an ancient city and its hinterland involving air, water, soils and sediments.

## Materials and methods

### Chronologies of urban pollution spreading: Radiocarbon AMS dating

All radiocarbon samples from the northwest quarter (NWQ) of Jerash are calibrated using the atmospheric calibration curve IntCal13 and the online calibration software Oxcal 4.2 [13, 14]. Human activity is inferred from summed calibrated probability distribution of all >150 radiocarbon $^{14}$C samples and kernel density estimation [14]. Significant periods of activity are estimated from summed probability distributions being higher than a 1000 randomly and uniformly distributed simulated dates with errors drawn from a probability distribution constructed from the actual 14C samples using Oxcal 4.2 and MatLab 2016 [15].

Fig 2A shows the human activity intensity within the city based on a radiocarbon cumulative probability plot of >150 radiocarbon analyses from the NWQ in Jerash [16]. The middle panel displays the summed probability distribution of all radiocarbon dates from Jerash Northwest Quarter. The black line represents inferred activity using kernel density estimation [14]. The top panel shows the significant summed probability distribution of all radiocarbon dates from Jerash NWQ. Peaks in this distribution indicate peaks in activity at the site, assuming unbiased sampling. The simple sum of radiocarbon dates could be biased by wiggles in the calibration curve, which could lead to peaks that were falsely interpreted as peaks in human activity. We calculated the significant summed probability distribution by subtracting the effect of the calibration curve from our summed dates in the following way: We took an average summed probability distribution of 1000 artificial (simulated) $^{14}$C samples indicating a situation where all $^{14}$C dates were equally distributed in time (randomly sampled). We subtracted this simulated sum distribution from the summed probability distribution of all our "real" samples.

### Chronologies of urban pollution spreading: OSL sediment chronologies

As part of the NWQ project, three profiles in the Upper, Middle and Lower reaches of the Wadi Suf were examined [18, 19]. These sections were sampled for OSL dating in 2015 and

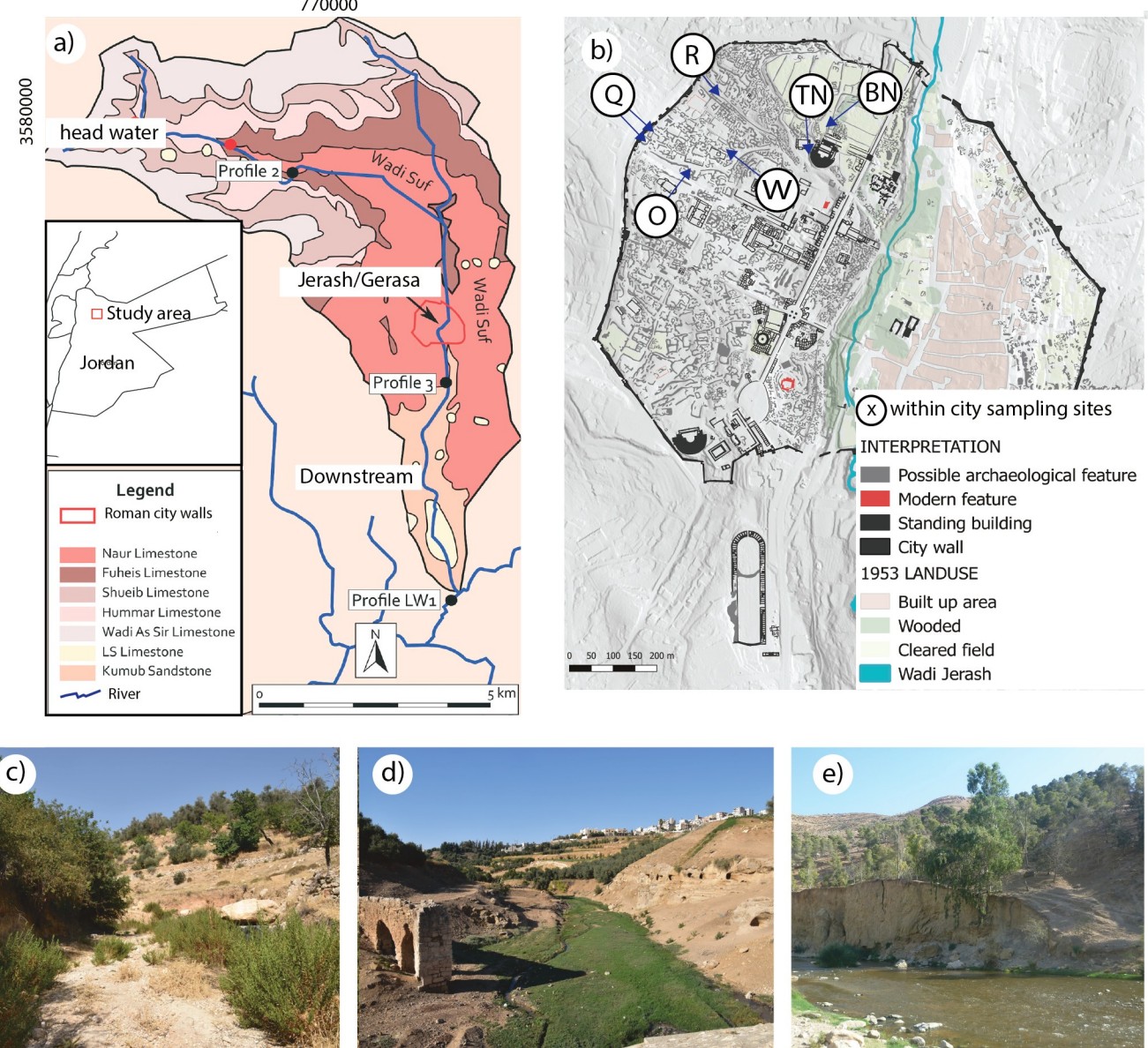

**Fig 1.** Map of the Wadi Suf watershed with the locations of profiles (A), map of the ancient city showing within city sampling locations (x), ancient sewage water sediments, Red Mediterranean soils, and urban sediments (B), and images of sampling locations in the wadi headwaters (C), the middle section (D) and downstream (E). Background maps from [11, 12].

2016, with the analyses conducted at SUERC (University of Glasgow) in early 2017. These early investigations showed that significant volumes of sediment were mobilized through the Wadi Suf from the mid 7th century CE to early 15th century CE, both prior to, and post the 749 CE earthquake. In 2017 we returned to the 2015–16 sections to retrospectively sample for OSL profiling, providing more detailed stratigraphy for each, using these relative chronologies to expand the investigations into the wider landscape [17, 20]. The bottom panel in Fig 2 shows the distribution of optically stimulated luminescence (OSL) depositional ages obtained for fluvial units in the wadi [17]. Evidence of fluvial sedimentation from the 2nd-3rd centuries BCE to the 7th century CE remained elusive, suggesting effective management of the water

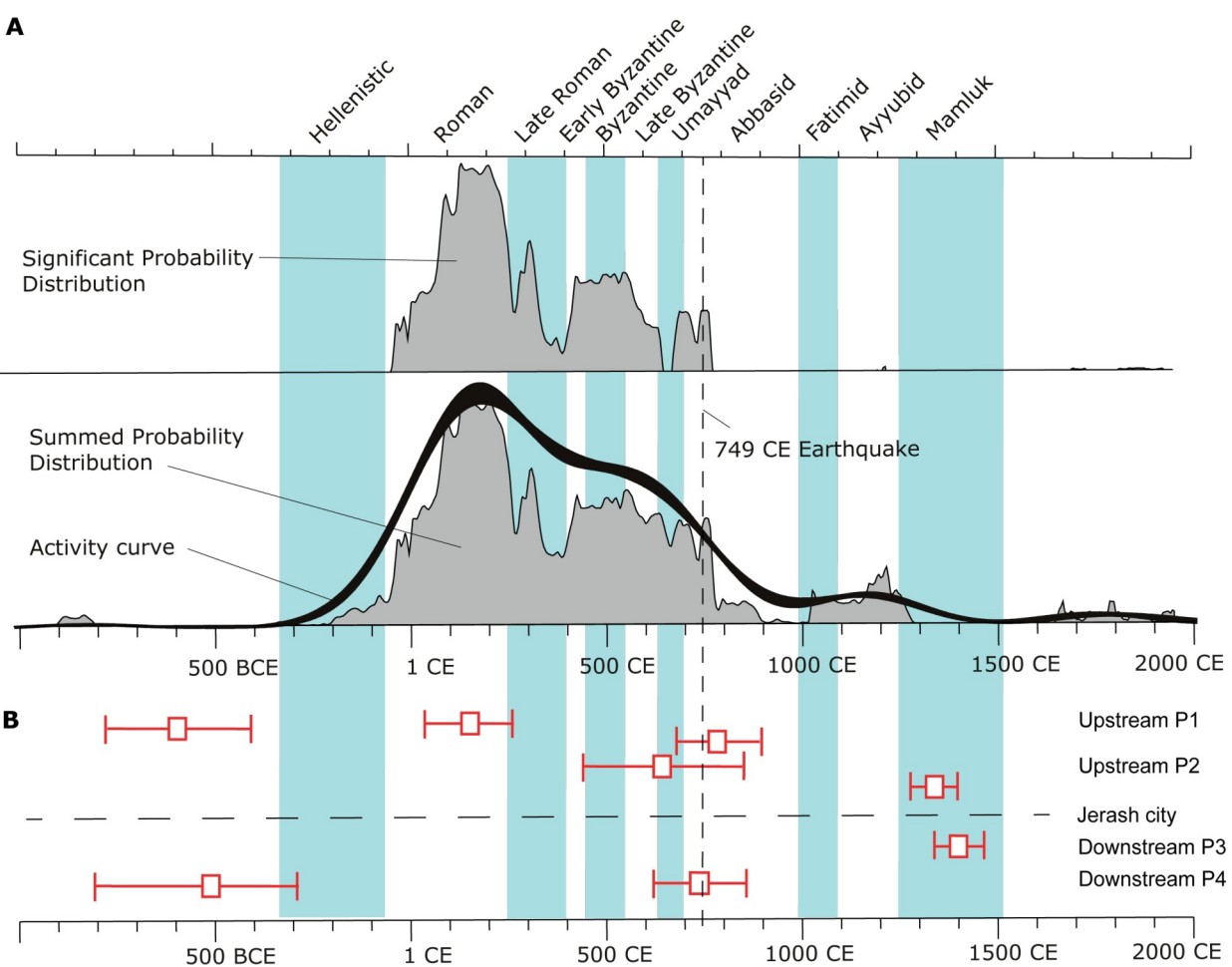

**Fig 2. Combined anthropogenic activity indicators.** Based on radiocarbon dating within the city (A) and estimates of sediment movement within the Wadi Suf based on optically stimulated luminescence (OSL) dating results (B). Data compiled from [16, 17].

and soil resources at this time. From the beginning of the 6th century, the wadi begins to in-fill with eroded soils, implying a decline in land and water management giving less resilience to associated drought conditions ca 500–750 CE and the later 749 CE earthquake impact [17].

## Elemental analyses of soils and sediments

Representative soil samples of ca. 250 g were collected from each stratigraphic context and pre-processed (2mm sieving and grinding) at the Department of Geoscience, Aarhus University. Element measurements were determined on representative subsamples, crushed fractions using strong acid digestion and Inductively Coupled Plasma-Mass Spectroscopy (ICP-MS) carried out by Acme Laboratories, Canada and University of Stirling, Scotland. An overview of analyzed elements can be found in the S1 Table, while selected anthropogenic pollutants can be seen in Fig S1 in S1 File.

## Ethical considerations

We thank the Department of Antiquities in Amman and Jerash for access to the site and artefacts.

## Results

We assess archaeological evidence (Supporting information) from Jerash to determine how and where enhanced heavy metal concentrations may have arisen, made possible through a high definition approach to excavation strategies in the Northwest Quarter (NWQ) of the ancient city [11, 21]. We then measure and compare heavy metal concentrations in two sets of chronologically controlled stratigraphies, one from within the city and the second from the Wadi Suf hinterland [17, 20, 22]. The spatial,temporal and contaminant contrasts between these stratigraphic units suggest local pathways of contaminated sediments, from sources to sinks.

Production based contaminant sources in the city include Roman through Umayyad artisanal metalworking activity delivering Cu and Pb to the urban sediments as evidenced by objects, metal slags and coinage made from leaded copper-based alloys [23, 24]. Copper coinage was also regularly minted in Jerash in the first to the third century CE and the seventh century CE [25]. Lead (e.g. in the form of minium, $Pb_3O_4$) and, to a smaller degree, Cu were incorporated into other materials, such as orange, green and blue pigments and used on late Roman through to Early Islamic wall paintings [26], while lead-glazed ceramics and leaded glass were rare and unlikely to have contributed significantly to elevated Pb. Hardly any lead pipes were observed in excavations or within the city's waste deposits [27].

Profiles examined in the ancient city (Fig 1) offer insight into urban public (a Basilica, BN) and private spaces (profile TW), with ages spanning the Roman through the Umayyad periods (Fig 2). The NT profile is associated with remnants of more recent agricultural terrace walls (possibly Mamluk or Circassian) and mainly Roman age material. Additionally, archaeological excavation allowed contamination loading assessments of remnant intact fragments of Red Mediterranean Soil underlying the urban NWQ (Area Q, O and R), Red Mediterranean Soil used as construction fill (Areas R, O and P) and sediments found in water and sewage channels (Area Q, O, P). Hinterland profiles are located in Wadi Suf, with sediment deposition OSL ages upstream spanning the Byzantine to Medieval periods, while deposition ages downstream of the city ranged from the Mamluk period to the mid-20th century.

Fig 2 illustrates that the within city activities in the NWQ of Jerash started during the first century BCE and virtually ceased after the 749 CE earthquake (2A), while the erosion and transport of the polluted urban sediments to the wadi, and their subsequent movement and redeposition within the wadi sediments downstream took place during the occupation and continued for centuries after its abandonment (2B). See also [16, 20].

Heavy metal contaminations of Cu, but especially Pb in profiles within the city (BN and NT), and midstream and downstream wadi profiles P2 and P3 (Figs 1 and 3) are notably higher than those of local bedrock and local and regional soils [28]. Here, elevated heavy metal values can be considered contamination, and in some samples reach soil pollution levels [12, 29]. Conversely, upstream wadi profiles have heavy metal values closer to the local geological background and are geochemically distinct from the Jerash NWQ and the lower wadi. The data (Fig 3, S1, S2 Tables) for within city soils and sediments indicate that the lowest values of Cu and Pb are within Profile TW, the domestic outdoor space. In contrast, higher values of Cu and Pb were observed within the lower strata of Profile BN, the outdoor communal space. The majority of high heavy metal values are associated with the Roman period (Supporting information) with lesser values from Byzantine through Umayyad periods. Heavy metal values of especially Pb fluctuate widely in the Late Roman material within Profile NT, while Cu exhibited a smaller range in values. Concentrations of Cu are highest in the sediments found in undisturbed mortar pipes from the city's sewerage system, while Pb levels on average are highest in disturbed sediments from within the city, and at their lowest concentrations in the Red

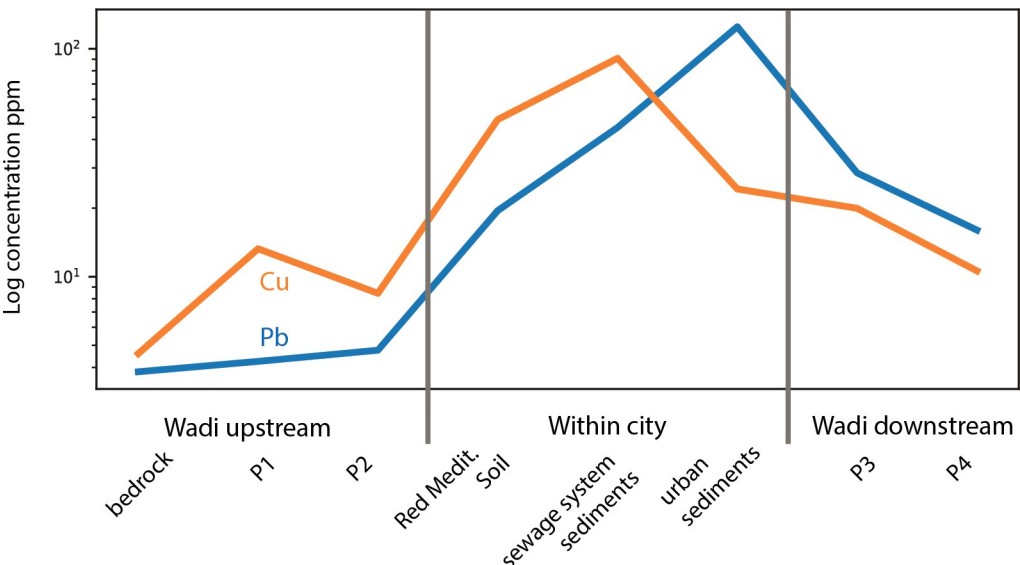

**Fig 3. Variations of metal values of Cu and Pb in soils and sediments upstream, within the city and downstream of the ancient city of Jerash**

Mediterranean soils underlying city structures. Lead and Cu values in wadi sediments demonstrate a distinct pattern of low heavy metal values in the upper wadi (P2), high values just below the urban area, and reduced, but still relatively high values in the lower wadi.

Elevated values of Pb and Cu observed in NWQ stratigraphy and wadi profiles downstream from the ancient city imply that metal use and metal-related activities were an important aspect of daily life. However, within the profiles, no direct, 'visible' evidence for metal-related industrial or artisanal activities (e.g., metal artifacts or metal-use related structures) is evident. Instead, the heavy metal concentrations of Pb together with Cu reflect materials introduced into the city with subsequent movement of contaminants by sediment transfer within the city and into the wadi (Fig 4).

## Discussion

### Archaeological evidence of urban heavy metal use in Jerash

Production based contaminant sources in the ancient city of Jerash include Roman and later metalworking activity evidenced by metal objects of mainly leaded Cu-based alloys with up to 32 wt% Pb and 11 wt% Sn but also Fe and Pb together with secondary production slags usually found in secondary contexts (e.g. [24, 30]). A bronze workshop installation associated with the sanctuary of Zeus and dated to the 2nd century CE gives further evidence of production based metalworking particularly with Cu in the Roman city [30, 31], while coinage was minted in Jerash regularly only in the first to the third century CE and for a very short period in the seventh century CE [32]. A recent study has demonstrated that Cu alloy coinage was leaded throughout the period from the 5th century CE until the middle of the 8th century CE [23]. This required the production of flans and therefore metal melting and working in the city. Small change coinage also circulated in the city from the 5th to at least the middle of the 8th century CE [23, 33]. Local gold smiths are attested from inscriptions of the Roman period [34]. Metal working would have led to primary accumulations of heavy metals in the urban sediments while later weathering of the produced metal-based materials and artefacts may contribute secondary contaminant sources in the city. Lead and copper have also been found to be incorporated into

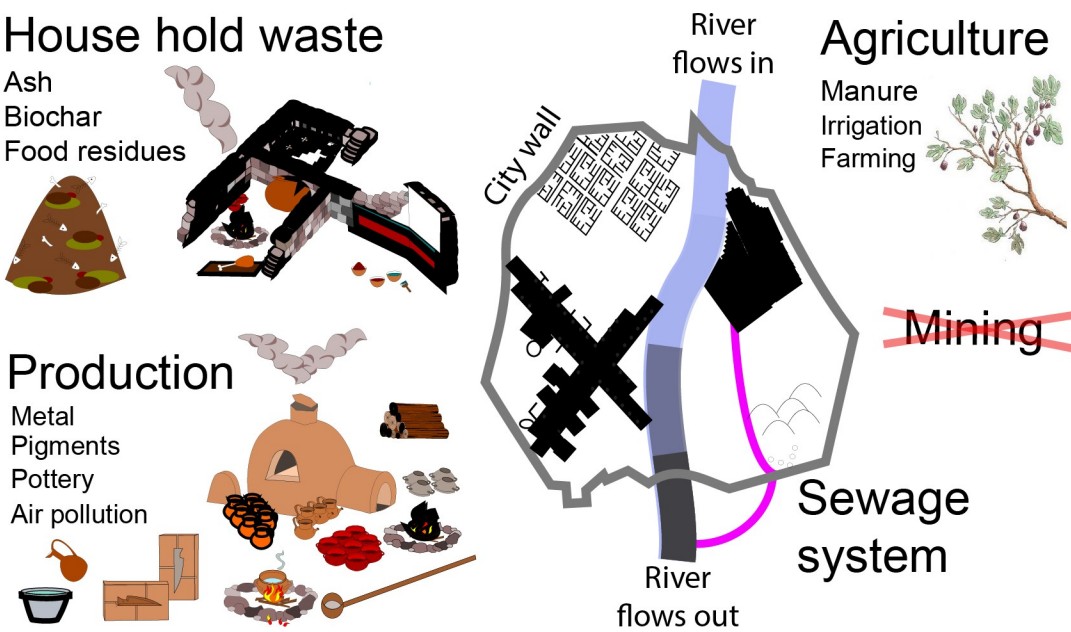

**Fig 4. A conceptual framework of major pollution sources associated with a Roman provincial city**

other materials, such as pigments in paint. Wall pigments as well as crucibles still containing pigment [26] have been dated to both the late Roman and Early Islamic period. These were found to include the orange pigment minium ($Pb_3O_4$), presumably produced locally from heating of the mineral hydrocerussite (also known as 'white lead' $PbCO_3xPb(OH)_2$) in air and, to a lesser degree, green and blue pigments derived from Cu-based minerals (see Table 2 in [26]). Although red pigments in wall paintings in ancient Jerash are generally dominated by the use of hematite ($Fe_2O_3$), they occasionally contain smaller amounts of arsenic compounds to intensify the red color presumably added in the form of the mineral realgar ($\alpha$-$As_4S_4$) [26].

Low Pb levels in translucent glass from the Hellenistic and Roman periods, with the majority of finds from the Byzantine periods varying from 3 (Hellenistic) to 100 ppm (Roman) are unlikely to have contributed significantly to Pb accumulation in the urban sediments [35], while a smaller secondary production of high-Pb colored glass (up to 40 wt% PbO) used for vessel decoration and tesserae for glass mosaics would have contributed some [35]. One surprising omission from the catalogue of contaminant sources in ancient Jerash is the rarity of archaeological evidence of Pb pipes, given that these have hardly been observed in the extensive excavations or within city waste deposits [27]. Metal mining and smelting sites are absent from the nearest 10s of km around Jerash, especially upstream. In summary, elevated Pb, Cu, Sn and As concentrations in the city's sediments correspond well with documented metal-related activity and the use of metal-rich wall pigments. Elevated silver concentrations in Jerash sediments, however, are more difficult to reconcile with urban activities given the near absence of silver objects in the excavated material culture. However, silver is present in copper and lead artefacts and given that silver is less soluble than copper and lead, it will tend to accumulate in the soils despite its lower overall presence in the artefacts.

## Contamination of the city and its hinterland wadi

Contaminated sediments moved with overland flow resulting in relocation of contaminants in outdoor spaces, such as within Profiles BN and NT. Surface erosion processes were primarily

responsible for the movement of sediments into, and through, Wadi Suf as contaminated sediments became incorporated into the sewage and waste-water system of the ancient city, flushed downstream [22]. Sediments presently stored in the wadi are primarily a legacy from Roman period urban activities and, to lesser degrees, the Byzantine through Umayyad periods (Fig 3). Atmospheric transfer is also evidenced as fine black particulate matter in the Red Mediterranean Soils within the city [12].

## Conclusions

Elevated levels of Pb and Cu in urban soils and sediments at ancient Jerash and its hinterland wadi result from the legacy of common, cumulative artisanal and daily activities, previously ignored in historical contamination studies. Local aeolian, fluvial, cultural and post-depositional processes elevating the levels of these contaminants in sediments downstream from Jerash' ancient city to form an 'anthropo-sequence'. They reflect long-term anthropogenic legacies at local scales, commencing in the Roman period. Later inhabitants unknowingly incorporated polluted sediments into their agricultural terraces that may have inadvertently affected the food and health of these later occupants as a consequence of urban contamination legacies. Long-term urban legacy effects also have implications for hinterland sediments and modern assessments of pollution, whereby heavy metal values in the wadi catchment may be used as a baseline to identify modern pollution or contamination indicators. The lack of a catchment scale waste management scheme and strict waste management during the Roman-Umayyad periods [10, 30] had unforeseen consequences for later occupants of the city and hinterland of Jerash and holds lessons for sustainable middle-sized cities of today and the future. As numerous cities of the size of Jerash and even larger existed in Antiquity, flourishing and expanding particularly from the first century CE onwards, it also urges global pollution studies, (e.g., [3, 36, 37]) to consider small-scale, daily-life use and reuse activities in urban settings as a potential important factors in understanding and explaining changes in the baseline of global pollution, not only today [38], but over the last millennia.

## Supporting information

**S1 Dataset.**
(XLSX)

**S1 Table.**
(PDF)

**S2 Table.**
(PDF)

**S1 File.**
(DOCX)

## Acknowledgments

Signe Kristensen and Thomas Ljungberg are acknowledged for help with illustrations. This research was undertaken within the framework of the Danish-German Jerash Northwest Quarter Project of the Universities of Aarhus and Münster, co-directed by Achim Lichtenberger and Rubina Raja. The project was undertaken between 2011 and 2017 with permission granted by the Department of Antiquities of Jordan, Amman before every campaign after application by the directors.

## Author Contributions

**Conceptualization:** Genevieve Holdridge, Søren M. Kristiansen, Achim Lichtenberger, Rubina Raja, Ian Simpson.

**Formal analysis:** Genevieve Holdridge, Søren M. Kristiansen, Gry H. Barfod, Tim C. Kinnaird, Achim Lichtenberger, Jesper Olsen, Bente Philippsen, Rubina Raja, Ian Simpson.

**Writing – original draft:** Genevieve Holdridge.

**Writing – review & editing:** Genevieve Holdridge, Søren M. Kristiansen, Gry H. Barfod, Tim C. Kinnaird, Achim Lichtenberger, Jesper Olsen, Bente Philippsen, Rubina Raja, Ian Simpson.

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
