## [Decision Letter · Decision Letter 0]

26 Mar 2021

PONE-D-21-01035

A Roman provincial city and its contamination legacy from artisanal and daily-life activities

PLOS ONE

Dear Dr. Kristiansen,

Thank you for submitting your manuscript to PLOS ONE. After careful consideration, we feel that it has merit but does not fully meet PLOS ONE’s publication criteria as it currently stands. Therefore, we invite you to submit a revised version of the manuscript that addresses the points raised during the review process.

One of the reviewers highlighted several points that you must address once revising your manuscript. All comments are aimed at increase the clarity of your manuscript and to better explain some of your assumptions.

We look forward to receiving your revised manuscript.

Kind regards,

Andrea Zerboni, Ph.D.

Academic Editor

PLOS ONE

Journal Requirements:

2. In your Methods section, please provide additional information regarding the permits you obtained to collect samples for the present study.

Please ensure you have included the full name of the authority that approved the field site access and, if no permits were required, a brief statement explaining why.

3. We note that Figure 1 in your submission contains map images which may be copyrighted.

We require you to either (a) present written permission from the copyright holder to publish this figure specifically under the CC BY 4.0 license, or (b) remove the figure from your submission:

b. If you are unable to obtain permission from the original copyright holder to publish this figure under the CC BY 4.0 license or if the copyright holder’s requirements are incompatible with the CC BY 4.0 license, please either i) remove the figure or ii) supply a replacement figure that complies with the CC BY 4.0 license. Please check copyright information on all replacement figures and update the figure caption with source information. If applicable, please specify in the figure caption text when a figure is similar but not identical to the original image and is therefore for illustrative purposes only.

Reviewers' comments:

Reviewer's Responses to Questions

**Comments to the Author**

1. Is the manuscript technically sound, and do the data support the conclusions?

Reviewer #1: Partly

Reviewer #2: Yes

2. Has the statistical analysis been performed appropriately and rigorously? 

Reviewer #1: Yes

Reviewer #2: Yes

3. Have the authors made all data underlying the findings in their manuscript fully available?

Reviewer #1: Yes

Reviewer #2: Yes

4. Is the manuscript presented in an intelligible fashion and written in standard English?

Reviewer #1: Yes

Reviewer #2: Yes

5. Review Comments to the Author

Reviewer #1: Overall: This is a well prepared manuscript which has explored the social-technological-environmental relationships in the ancient city of Jerash. The data presented are interesting but there are questions in the way the results are interpreted. A major concern is with the sources of metals to account for the large-scale contamination of Gerasa/Jerash and its hinterlands. The level of heavy metal loading into the environment is clearly out of balance with what would be expected from “small-scale but common Roman urban, artisanal and everyday site activities”. The authors claim that the main uses of metals in the city included artisanal metalworking activity, production of copper coinage and making of pigments. The environmental footprints of such activities in all likelihood were highly localized within the residential areas. The total number of people involved in such activities was probably small considering that Jerash was a famous trading center and was not particularly known for its meta-working prowess. The question of where did all the metal pollution come from was not answered satisfactorily in the paper. The authors should consider the possibility that there was a lost lead/gold or lead/silver mine in or near the city which had released enough metalliferous wastes and fumes to contaminate the entire city and its neighborhood.

Comment 1 (Lines 119-123): The documentation of the source-sink relationships between the concentrations of metals in urban soils and Wadi Suf sediments is weak. The following statements (lines 119-123) support this concern: “Evidence of fluvial sedimentation from the 2nd-3rd centuries BC to the 7th century CE remained elusive, suggesting effective management of the water and soil resources at this time. From the beginning of the 6th century, the wadi begins to in-fill, and in the absence of additional chronological data may imply a decline in land and water management with associated erosion of soils to the wadi”. It should be noted that the graphs (of Cu and Pb concentrations at the specific locations) in Figure 3 do not include the time information for the samples. The peaks in the graph should be interpreted with caution unless the samples are not of the same age.

Comment 2: The role of local climate conditions has not been adequately addressed. There are reports suggesting that the late Roman Empire period was marked by major climatic changes in the Middle East which could have had some influence in the supply and accumulation of sediments in the river basin. A major floor can easily mediate the amount and type of sediments deposited at any location in the Wadi Sufa basin. Interpreting the sedimentary record solely on the basis of human activities (without consideration of the geophysical and geochemical processes) is an over-simplification.

Comment 3 (Figure 2): What is the error range in the stimulated results?

Comment 4: The following statement (lines 138-141) cannot be justified: “We then measure and compare heavy metal concentrations in two sets of chronologically controlled stratigraphies, from within the city and from the Wadi Suf hinterland (13-15). The spatial and temporal relationships between these stratigraphic units enable assessment of local pathways, from sources to sinks”. It would be an impossible task (almost) to reliably infer the transfer pathways for the metals in the particular environment based on their concentrations in such divergent locations. In the first instance, lead is immobile in soils and the mechanism for its transfer from within the city to the Wadi Suf hinterland needs to be explained.

Comment 5 (Figures 5, S6 and S7): It is difficult to infer any historical relationships in these figures without showing any time lines. These figures probably should be redone.

Comment 6 (lines 221-222): What is the evidence that the weathering of metal-based materials and artefacts represents “significant secondary sources” of environmental metal contamination in the city? Even if so, the impacts would be very highly localized.

Comment 7 (lines 244-248): The statement that elevated silver concentrations of Jerash sediments came from secondary sources such as Pb-rich material since some galena ores contain significant amounts of silver (argentiferous galena) is interesting. It hints at a likelihood that lead ores might have been exploited in the area in ancient times. The lead deposits would have been exhausted by now.

Comment 8 (Figure S5): The difference in the spread of Pb and Cu concentrations in on-site versus off-site sediments needs to be explained

Comment 9 (Conclusions): The study purportedly shows that most of the lead and copper released into the environment in Jerash city are deposited locally. There is no evidence to suggest that the metal works in the city were able to release much of the metals into the atmosphere for regional and global circulation. The conclusion that “small-scale, daily-life use and reuse activities in urban settings are an important factor in understanding and explaining changes in the baseline of global pollution, not only today, but over the last millennia” is misleading.

Reviewer #2: The data, methodologies, and conclusions are all clearly presented and discussed. The findings are significant for the archaeological community and for other urban studies disciplines. I endorse publication.

6. PLOS authors have the option to publish the peer review history of their article (what does this mean?). If published, this will include your full peer review and any attached files.

Reviewer #1: No

Reviewer #2: No

---

## [Author Response · Author response to Decision Letter 0]

30 Apr 2021

Please see attached file with REsponse to Reviewers"

---

## [Decision Letter · Decision Letter 1]

6 May 2021

A Roman provincial city and its contamination legacy from artisanal and daily-life activities

PONE-D-21-01035R1

Dear Dr. Kristiansen,

We’re pleased to inform you that your manuscript has been judged scientifically suitable for publication and will be formally accepted for publication once it meets all outstanding technical requirements.

Kind regards,

Andrea Zerboni, Ph.D.

Academic Editor

PLOS ONE

Additional Editor Comments (optional):

Reviewers' comments:

Reviewer's Responses to Questions

**Comments to the Author**

1. If the authors have adequately addressed your comments raised in a previous round of review and you feel that this manuscript is now acceptable for publication, you may indicate that here to bypass the “Comments to the Author” section, enter your conflict of interest statement in the “Confidential to Editor” section, and submit your "Accept" recommendation.

Reviewer #1: All comments have been addressed

2. Is the manuscript technically sound, and do the data support the conclusions?

Reviewer #1: Yes

3. Has the statistical analysis been performed appropriately and rigorously? 

Reviewer #1: Yes

4. Have the authors made all data underlying the findings in their manuscript fully available?

Reviewer #1: Yes

5. Is the manuscript presented in an intelligible fashion and written in standard English?

Reviewer #1: Yes

6. Review Comments to the Author

Reviewer #1: It is doubtful that the sources mentioned in the paper can account for the substantial amounts of lead contamination in Jerash and its surrounding areas. The paper is very well written and presents interesting data on paleo-pollution which should be available in the scientific literature. I recommend publication of the paper in its present form.

7. PLOS authors have the option to publish the peer review history of their article (what does this mean?). If published, this will include your full peer review and any attached files.

Reviewer #1: **Yes: **Jerome Nriagu

---

## [Editor Report · Acceptance letter]

24 May 2021

PONE-D-21-01035R1 

A Roman provincial city and its contamination legacy from artisanal and daily-life activities 

Dear Dr. Kristiansen:

I'm pleased to inform you that your manuscript has been deemed suitable for publication in PLOS ONE. Congratulations! Your manuscript is now with our production department. 

Kind regards, 

on behalf of

Prof. Andrea Zerboni 

Academic Editor

PLOS ONE